# The Status of Molecular Analyses of Isolates of *Acanthamoeba* Maintained by International Culture Collections

**DOI:** 10.3390/microorganisms11020295

**Published:** 2023-01-23

**Authors:** Paul A. Fuerst

**Affiliations:** Department of Evolution, Ecology and Organismal Biology, The Ohio State University, Columbus, OH 43210, USA; fuerst.1@osu.edu

**Keywords:** *Acanthamoeba*, ATCC, CCAP, BEI, 18S rRNA, 16S-like rRNA, Sequence Types, culture collections, standard strains

## Abstract

*Acanthamoeba* is among the most ubiquitous protistan groups in nature. Knowledge of the biological diversity of *Acanthamoeba* comes in part from the use of strains maintained by the major microbial culture collections, ATCC and CCAP. Standard strains are vital to ensure the comparability of research. The diversity of standard strains of *Acanthamoeba* in the culture collections is reviewed, emphasizing the extent of genotypic studies based on DNA sequencing of the small subunit ribosomal RNA from the nucleus (18S rRNA gene; *Rns*) or the mitochondria (16S-like rRNA gene; *rns*). Over 170 different strains have been maintained at some time by culture centers. DNA sequence information is available for more than 70% of these strains. Determination of the genotypic classification of standard strains within the genus indicates that frequencies of types within culture collections only roughly mirror that from clinical or environmental studies, with significant differences in the frequency of some genotypes. Culture collections include the type of isolate from almost all named species of *Acanthamoeba*, allowing an evaluation of the validity of species designations. Multiple species are found to share the same Sequence Type, while multiple Sequence Types have been identified for different strains that share the same species name. Issues of sequence reliability and the possibility that a small number of standard strains have been mislabeled when studied are also examined, leading to potential problems for comparative analyses. It is important that all species have reliable genotype designations. The culture collections should be encouraged to assist in completing the molecular inventory of standard strains, while workers in the *Acanthamoeba* research community should endeavor to ensure that strains representative of genotypes that are missing from the culture collection are provided to the culture centers for preservation.

## 1. Introduction

Individuals making up the genus *Acanthamoeba* are among the most prevalent protists in the environment. They have been found in a great variety of environments, usually associated with biofilms, where they act as predators of bacteria and other single-celled eukaryotes [1]. In terrestrial systems, they have been easily isolated from soil and dust [2]. They are also found routinely in a wide assortment of aquatic environments. Salinity seems of only limited importance, since fresh water, brackish water, and sea water have all yielded isolates [3,4]. Among the varied aquatic setting that have yielded samples are sewage, swimming pools, contact lens solutions, air conditioning systems, and clinical equipment such as medicinal pools, dental treatment units, and dialysis units, and even from emergency eye wash stations [5], an unfortunate occurrence since the medical condition most associated with *Acanthamoeba* is the eye disease *Acanthamoeba* keratitis (AK). *Acanthamoeba* can often be found contaminating other cell cultures. The original standard strain of *Acanthamoeba* was isolated by Castellani [6] as a contaminant of a yeast culture. *Acanthamoeba* may also occur as a contaminant on food, such as vegetables, presumably transferred from soil, although *Acanthamoeba* is not usually thought of as a major source of intestinal disease. In humans and other mammals such as dogs, *Acanthamoeba* is often isolated from nasal or throat swabs. Several studies of the prevalence of antibodies against *Acanthamoeba* in humans suggest that exposure to *Acanthamoeba* is very common, even routine [7,8,9,10].

Up to 30 species of *Acanthamoeba* have been described, based on morphological and/or molecular criteria [11,12,13,14]. Various species of *Acanthamoeba* have been implicated as the cause of several different infections of vertebrates. These diseases can range from infections of the skin, through more frequent infections of the eye (*Acanthamoeba* keratitis, AK), to less frequent but very serious and potentially lethal infections of the brain (granulomatous amoebic encephalitis, GAE), or even to diffuse multiorgan infiltration [15]. Each of these manifestations except AK is very rare in immunocompetent individuals. It is clear that *Acanthamoeba* is growing as a recognized threat to health around the world, even if we restrict our focus to only the role of these amoebae in the severe sight-threatening ocular infection AK [16,17].

Finally, the genus is also characterized by its ability to sustain, and in many cases maintain, a variety of obligate or facultative intracellular bacteria, many of which are known pathogens [18,19,20,21,22]. Members of *Acanthamoeba* are therefore potential vectors for many highly pathogenic prokaryotes, in part shielding these bacteria from antibacterial treatments.

## 2. The International Stock Centers

To better understand how *Acanthamoeba* lives, reproduces, and causes disease, laboratory studies on the biology of *Acanthamoeba* are vitally important. The use of standard stock strains in experiments becomes an important aspect of research, allowing evaluation and comparison between studies. In the study of *Acanthamoeba*, two stock centers have performed the main task of providing researchers with isolates of known provenance. These are the American Type Culture Collection (ATCC) and the Culture Collection of Algae and Protozoa (CCAP). Investigators are able to deposit material, especially cultures of isolates, that have been obtained in either environmental or clinical studies. The stock centers provide material for use by investigators interested in pursuing standardized studies. Variation in biological materials can affect reproducibility, and thus can bias the conclusions drawn from studies of *Acanthamoeba* in the environment or in the laboratory. The use of standard strains of organisms can help control for and counteract these effects.

Many strains of *Acanthamoeba* have been available from the two stock centers. The number of standard strains available has increased or decreased over time, dependent on the varied interest of workers in the field to deposit newly isolated strains and on the difficulty of maintaining amoebic material that requires intermittent specialized culture procedures. Nevertheless, the standard strains maintained by the culture centers can be used to provide the baseline for the analysis of genetic variation within the genus *Acanthamoeba*.

The material represented by culture collections is especially important for the understanding of diversity within the genus *Acanthamoeba*. As mentioned, 30 or more species have been described within the genus *Acanthamoeba* [11,12,23,24]. The culture collections have been the repository for isolates that represent most of the type specimens for newly or provisionally named species. For *Acanthamoeba*, this role takes on increased importance because DNA data suggest that species designations may not reflect the phylogenetic relationships between strains. Phylogenetically identifiable clades may include redundant classifications [12,13,25,26,27]. Analysis of DNA similarity between the isolates maintained in the culture collections can help to assess the legitimacy of species designations and suggest how species designations can be clarified in *Acanthamoeba*.

Both ATCC and CCAP now use online catalogs to provide information about strains. These catalogs have themselves evolved over time, changing the background information that is available to investigators. Examination of the catalog information from either stock center, and comparisons with earlier versions of the catalogs indicate that the inventories have changed over time and that the information considered vital by the stock center does not always provide unequivocal information concerning the strains being maintained.

Genetic studies to clarify and classify the relationship among strains of *Acanthamoeba* have progressed greatly over the last 25 years [26,28,29]. These results have only been incorporated sporadically into the information that the culture centers maintain on the standard strains. Here, the enumeration of the standard strains of *Acanthamoeba* that are, or have been, available from the stock center is presented, and a census is taken to determine whether genetic information has been collected for these strains. The international DNA databases have been examined to determine whether DNA sequences have been obtained for strains listed in the inventory of the two stock centers. We have emphasized determining DNA sequences from either of two genes that have been major foci of phylogenetic analysis in *Acanthamoeba*, the nuclear 18S rRNA gene, and the mitochondrial 16S-like rRNA gene. In cases where a strain has not been studied using DNA sequences, we have tried to determine whether alternative information exists. Alternatives that we considered included RFLP patterns for the mitochondrial genome of a strain [30,31,32,33] or standardized allozyme patterns obtained by protein electrophoresis [25].

## 3. How Many Standard Strains Exist?

The online catalogs of the two stock centers were surveyed to determine the number of standard strains of *Acanthamoeba* or closely related genera that were available in Autumn 2022. In addition, information from catalogs of CCAP from 1976, 1982, and 2001 was examined to determine whether there are strains that might have been available previously but were no longer maintained for distribution. Similar previous catalogs for ATCC were available from 1970, 1972, and 1993, and staff at ATCC generously provided information concerning a number of possible strains that were unavailable or have been discontinued.

Of the two stock centers, ATCC is clearly the dominant source of material on *Acanthamoeba*. ATCC currently lists 162 cultures of *Acanthamoeba* in its online catalog (Table 1). At least 11 of the ATCC cultures represent duplicate submissions of an isolate (four original isolates each represented by two ATCC listings, and one isolate represented three times). The duplicate listings represent cultures submitted by investigators to certify the identity of an isolate used in a study or represent an axenic culture of a previous submission. Additionally, there is a single standard culture of *Comandonia operculatum* which has been shown genetically to be included within *Acanthamoeba* [34]. While *Comandonia* was initially described morphologically as a genus distinct from *Acanthamoeba* [35], the genetic data indicate that the isolate identified as *C. operculatum* and held by ATCC falls clearly within *Acanthamoeba*. It is possible, however, that other amoebae identified morphologically as *Comandonia* could represent a distinct genus, possibly more related to *Flamella* [36]. With respect to other Amoebozoan genera considered to be members of the Acanthamoebidae, ATCC also maintains three cultures of *Balamuthia mandrillaris* and a single culture of *Protacanthamoeba caledonica.* CCAP does not maintain any isolates of *B. mandrillaris*, and no longer maintains a stock of *P. caledonica*.

An additional aspect of the ATCC collection of isolates requires consideration. ATCC maintains samples in two categories. The first category includes those isolates that are immediately available to researchers, although fees and restrictions on use vary. The second category of strains are those referred to as “Mission Collection Items”. Mission collection strains are low in stock, and not part of the standard ATCC inventory. A special production run is required to produce samples. Such isolates are provided on a customer-requested basis only, with a cost that may be 10–20× that for non-Mission strains. Examples of the isolates that fall into these fees/restriction classes include the two alternative versions of the Neff strain at ATCC. The product with identification ATCC 30010 is available readily from the culture center, while ATCC 50373, a resubmission of the strain after a determination of its 18S rRNA gene sequence [28], is classified as a Mission Collection isolate.

Other examples of potentially important strains classified as Mission Collection isolates include the type specimens for at least nine nominal species of *Acanthamoeba*. These include ATCC 30135 (*A. comandoni*), ATCC 50239 (*A. echinulata*), ATCC 30867 (*A. tubiashi*), ATCC 50238 (*A. divoniensis*), ATCC 50240 (*A. lugdunensis*), ATCC 30134 (*A. terricola*), ATCC 30866 (*A. healyi*), ATCC 30732 (*A. jacobsi*), and ATCC 30870 (*A. palestinensis*). Mission Collection strains represent 69 of the 163 *Acanthamoeba/Comandonia* isolates in the current ATCC catalog.

ATCC also manages a second source of cultures for the scientific community, BEI Resources. BEI Resources was established by the National Institute of Allergy and Infectious Diseases (NIAID) to provide reagents, tools, and information for studying Category A, B, and C priority pathogens, emerging infectious disease agents, non-pathogenic microbes, and other microbiological materials of relevance to the research community. BEI Resources maintains material from 27 isolates of *Acanthamoeba*, 15 of which are also available directly through ATCC, resulting in 175 total ATCC associated active *Acanthamoeba* cultures (Table 1). BEI Resources also maintains six isolates of *B. mandrillaris*, yielding 9 total ATCC associated *Balamuthia* cultures.

In addition to the active isolate cultures listed in the ATCC or BEI catalogs, at least nine strains have existed to which ATCC numbers have been assigned in the past. Eight of these isolates have appeared in the scientific literature, having been used in various studies on *Acanthamoeba*. Two of these strains with putative inactive ATCC identifiers are available in the BEI catalog, but do not appear to be currently available from the ATCC collections. One strain was reported to us with an ATCC identifier, but no genetic information or literature reference to the strain has been found. We do not believe that any other isolates previously available from ATCC have been discontinued, although this is not absolutely certain, since older ATCC catalogs between 1999 and present were not easily available without major effort by the ATCC staff. However, we have not ascertained any other ATCC numbers among the list of isolates used in studies in the literature.

In contrast to the large number of ATCC cultures, the online catalog of CCAP currently lists 23 strains of *Acanthamoeba*. A comparison of the catalogues of the two stock centers indicates considerable overlap. Thirteen of the isolates that are maintained by CCAP are shared with the ATCC, in part because some of the available CCAP cultures represent older North American isolates that appear to have been originally deposited in the ATCC collections. No cultures of *Balamuthia* or *Protacanthamoeba* are currently maintained by CCAP, but CCAP previously maintained the type culture of *P. caledonica* that is still available from ATCC. Examination of earlier CCAP catalogs indicates that at least 8 *Acanthamoeba* cultures once available through CCAP are no longer being maintained in the CCAP collection. Some of these, however, were shared with ATCC and still appear in the ATCC catalog, although often without acknowledging the former CCAP designation. Examination of information about the history of an isolate allows a cross reference between the two centers to be made for these discontinued/unavailable cultures, showing that 5 of the 8 unlisted *Acanthamoeba* CCAP isolates are still maintained by ATCC.

Among those cultures maintained by CCAP but also listed in the ATCC collections, one isolate, CCAP 1501/3c, has a problematic status. Information on the origin of CCAP 1501/3c shown in the catalogs of the two culture collections, and information from DNA sequences that has been determined, suggest that the isolate in ATCC (*A. palestinensis* 2802, ATCC 50708), despite being listed as CCAP 1501/3c, is not the same as the original CCAP 1501/3c isolate (OX-1).

Finally, some isolates that have been used in research on the biology of *Acanthamoeba* are no longer maintained in either of the stock center collections or are currently unavailable for distribution (Table 1). These are isolates that have been used in the past but are currently not listed in the online catalogs of either stock center.

In summary, as of late 2022, the two culture collections combined could provide material representing 180 *Acanthamoeba* isolates for study. Ten additional discontinued or unavailable cultures have also been identified, six of which have been used in research.

## 4. Linking Culture Center Isolates with Genetic Information

To clarify how much information about the standard strains has accumulated, we examined the international DNA databases. DNA sequences that have been deposited in the international DNA databases were examined using GenBank [37] as the primary resource, since the three major DNA databases (EMBL, DDBJ, and GenBank) maintain uniformity by exchanging information on a regular basis [38]. Our laboratory has been steadily maintaining updates concerning DNA information on all isolates of *Acanthamoeba* for which molecular studies of two phylogenetically informative genes have been reported. These are the ribosomal small subunit rRNA genes of the nucleus (18S rRNA gene) and mitochondria (16S-like rRNA gene). These represent the two genes that have been the focus for phylogenetic analysis in *Acanthamoeba* [29].

### 4.1. Small Subunit rRNA Genes

A summary of the information in the DNA databases, together with additional unpublished or undeposited sequence information provided by multiple investigators, showed that in the middle of 2014 over 330 almost complete 18S rRNA gene sequences (sequences exceeding 2000 nucleotides in length) and about 1500 partial sequences of the gene had been made available for analysis [29]. These numbers have increased substantially since 2014, resulting in more than 6100 complete or partial sequences in Spring 2022 (see the website: http://u.osu.edu/*acanthamoeba*/ accessed on 1 July 2022). This collection of sequences was inspected to determine whether sequences for either gene had been obtained for a particular ATCC or CCAP standard strain.

In addition to the nuclear genes, over 200 sequences of the mitochondrial 16S-like rRNA are also available in the DNA databases, some of which have been extracted from genomic data deposited in the databases.

### 4.2. Connecting Isolate Information with the DNA Databases

To link DNA data with the standard strain, we first determined how many of the standard strain isolates have been examined by DNA sequencing of either the nuclear 18S rRNA gene (*Rns*) or the mitochondrial 16S-like rRNA gene (*rns*). Sequences of *Acanthamoeba* rRNA genes that have been deposited in the DNA databases were cross-referenced with the standard strains of the culture centers. Some additional undeposited sequences have been made available to us from collaborating researchers as part of an ongoing effort to better understand the phylogeny of *Acanthamoeba*. The summary breakdown for the isolates from the culture centers is given in Table 1. At least a partial DNA sequence from one of the two rRNA genes has been determined for 151 strains of the 190 possible standard strains (including discontinued/unavailable strains) from the culture centers. For the remaining 39 strains, 14 have been studied either by using DNA RFLP analysis or by using allozymes. There were 25 standard cultures for which no DNA sequences or related molecular information has been gathered. The single *Comandonia* culture maintained by ATCC was folded into the analysis as a member of *Acanthamoeba.* It has been studied for the sequence of the 18S rRNA gene. Included in this analysis are a set of 5 isolates of *Acanthamoeba* that our analysis suggests were issued ATCC ID numbers, but for which there is no clear record that the isolates were ever deposited in the collections of the ATCC.

Details concerning specific ATCC or CCAP strains are provided in tables in the appendix. Information in these table includes the species designation for the strain as listed in the culture center catalogs, the strain name or designation, designations of the strain if maintained in multiple culture center listings, and a listing of the SSU rRNA gene sequences associated with the particular isolate. The gene sequence information includes GenBank accession numbers for the sequences for each strain, length of sequence, and Sequence Type designation assessed by the 18S rRNA gene sequence as developed originally [28] and subsequently expanded. Mission Isolate status for various ATCC strains is also provided. Appendix A provides details on ATCC strains currently in culture for which sequences from one or both of the SSU rRNA genes have been determined. Appendix A provide similar details for CCAP strains currently in culture. Appendix A list details for strains not currently available from a culture collection but for which molecular information has been collected and presented in the literature. Appendix A provides information for strains available from BEI for which sequences have been determined. Appendix A provides information on strains for which only allozyme or RFLP information is known to have been collected. Appendix A provides a list of strains for which no molecular data are currently available. Appendix A provides information on the culture center isolates of *Balamuthia* and *Protacanthamoeba*. Appendix A lists inactive cultures for which genetic information is lacking. Reference sources for the sequences of all isolates are provided in Appendix A.

As shown in Table 1, the largest group of isolates represents those that have been studied only for the nuclear 18S rRNA gene. About half as many isolates have been examined using both the nuclear and mitochondrial small subunit rRNA genes.

### 4.3. Genome Projects for Isolates of the Culture Center in the DNA Databases

Some of the information on small subunit rRNA genes has been extracted from information gathered for genome projects that have examined isolates of *Acanthamoeba*. No less than 33 genome projects have been focused on various strains of *Acanthamoeba*. At least five additional projects studied complete mitochondrial genomes of isolates. Of the isolates that have been studied as part of genome projects, 19 of these isolates have been maintained in the culture centers. The culture center isolates for which genome project information is available are shown in Table 2.

As we pointed out in an earlier paper [12], several of the projects originally involved erroneous labeling of the source of the isolates. The information in Table 2 provides our best understanding of true source of the data. An additional genome project, CDEZ, putatively represented A. royreba ATCC 30884. However, none of the gene sequences extracted from the genome project match those of A. royreba, nor do they match sequences that have been reported for any isolate of Acanthamoeba. They have thus been assigned to a separate Sequence Type T22 [12]. The fact that the genome information for Sequence Type T22 was part of a multi-isolate genome study, all isolates of which were meant to represent culture center standard strains, suggests the possibility that T22 may represent the information from one of the ATCC isolates for which no genetic information is yet available.

## 5. How Reliable Is the Sequence Information for Standard Strains of Acanthamoeba?

Given the importance of standard strains to the understanding of the phylogenetic relationships between the Sequence Types within the genus, and to the proper designation of species, it is appropriate to question whether the sequences that are currently available from standard strains can be used to provide a picture of evolution within Acanthamoeba that is accurate and reliable. There are two aspects of reliability: reliability from the viewpoint of sequencing accuracy and reliability from a phylogenetic viewpoint.

### 5.1. Assessing the Accuracy of Sequences in the DNA Database

To investigate the issue of sequence accuracy, we examined the sequence information that has been collected for evidence of any significant discordance in sequences. One way to determine sequence accuracy utilizes the fact that for a subset of 36 isolates the 18S rRNA gene has been sequenced independently by several investigators, yielding DNA comparisons that can be assessed for consistency and accuracy. Among this subset of strains are some of the isolates that are considered the most important for understanding the biology of Acanthamoeba. This is a principal reason that they have been sequenced repeatedly. A further subset of 14 isolates have been sequenced more than twice, including the Neff strain (independently sequenced 12 times under several different entries (ATCC 30010, ATCC 50373, and CCAP 1501/1a or 1501/1b), and the original A. castellanii (Ac30) strain (sequenced six times under entries ATCC 30011, 30234, 50374, CCAP 1501/2a or 1501/10). In total, the 36 independent standard strains have yielded 106 sequences that can be used for pairwise comparisons.

Concerning 16S-like rRNA gene sequences, twelve isolates include multiple sequences that have been determined independently. Three isolates, again including the Neff strain, have more than two 16S-like rRNA sequences in the databases. Collectively, multiple studies yielded 29 sequences, encompassing a total of 25 pairwise comparisons for the 16S-like rRNA gene.

### 5.2. Accuracy of 18S Sequences in the DNA Database

For each combination of isolate and gene, duplicate sequences were aligned, and the sequence similarity was determined. The results for the 18S rRNA gene are shown in Figure 1. Several outcomes of comparisons can be differentiated in the pairwise comparisons. First, the results summarized overall in Figure 1 represent a total of 160 pairwise comparisons. Results from the Neff isolate are especially instructive about the quality of sequence comparisons, and the possibility of isolate misidentification or mislabeling. The Neff isolate is usually listed as A. castellanii Neff, but recent analyses suggest that it should be classified as A. terricola Neff [13]. In Figure 1A, results of comparisons among 11 independent sequences obtained for the Neff isolate are shown. All comparisons showed pairwise similarity over 0.995, with 27 of the 55 pairwise comparisons showing complete identity. Among the 11 Neff sequences are six that span the total or near total length of the 18S rRNA. None of these sequences are exactly the same, having between two and six differences over ~2300 bases of comparison. A twelfth isolate listed as representing ATCC 30010 in a publication on Acanthamoeba encystment [39], (GenBank accession # EF554328), does not appear to be equivalent to other Neff sequences. The results of the pairwise comparison of this isolate to the remaining 11 Neff sequences are shown in Figure 1B. All comparisons with this sequence show sequence similarity below 0.985. Differences between EF554328 and other Neff sequences occur primarily in three regions of the gene that show increased interstrain variability within Acanthamoeba. Examination of characteristic JDP1-JDP2 sequence motifs in the hypervariable regions of the 18S rRNA gene [40] make it very unlikely that this sequence represents the Neff strain. It appears that the isolate used in the encystment study was mislabeled as ATCC 30010. EF554328 is not identical to any other standard strain, differing by at least 7 changes from the closest other standard isolate, ATCC 50493.

Analysis of another set of Acanthamoeba early sequences illustrates the possibility of sequencing errors in sequences deposited in the DNA databases. Comparison with other GenBank sequences was made using results from a study that included a number of standard strains [41]. The study by Khan, Jarroll, and Paget included 7 standard strains. These sequences produced 17 pairwise comparisons when aligned to sequences obtained in other studies. Results are shown in Figure 1C. A contrast can be made to the matches between sequences from other studies that are shown in Figure 1D. Whereas only two of the contrasts in Figure 1C have similarities above 0.990, 62 of 77 pairwise tests exceed 99% similarity. Three contrasts in Figure 1C have below 0.95 similarity with equivalent sequences, while no comparison in Figure 1D is that low. Further, however, the 7 sequences from Kahn, Jarroll, and Paget all share similar multiple insertion/deletion differences from other Acanthamoeba sequences in the DNA databases, suggesting that the divergences were due to sequencing artifacts specific to that early study. When sequence comparisons were restricted to only the critical JDP1-JDP2 motif region of the 18S rRNA gene, 6 of the 7 sequences from Khan, Jarroll, and Paget were identical with the matched sample. Only a single sample, AF239298, representing CCAP 1501/3c (OX-1) showed differences. This last sample was at least 5% divergent from any other sequence within the DNA databases for the critical JDP1-JDP2 region and was the most divergent sample among all matched samples tested. Nevertheless, despite questions about the sequence, the placement of AF239298 within the phylogeny of Acanthamoeba is still within the super-type T2/T6, consistent with the placement of the other sample of CCAP 1501/3c (Genbank AF019051).

Analysis of another set of Acanthamoeba early sequences illustrates the possibility of sequencing errors in sequences deposited in the DNA databases. Comparison with other GenBank sequences was made using results from a study that included a number of standard strains [41]. The study by Khan, Jarroll, and Paget included 7 standard strains. These sequences produced 17 pairwise comparisons when aligned to sequences obtained in other studies. Results are shown in Figure 1C. A contrast can be made to the matches between sequences from other studies that are shown in Figure 1D. Whereas only two of the contrasts in Figure 1C have similarities above 0.990, 62 of 77 pairwise tests exceed 99% similarity. Three contrasts in Figure 1C have below 0.95 similarity with equivalent sequences, while no comparison in Figure 1D is that low. Further, however, the 7 sequences from Kahn, Jarroll, and Paget all share similar multiple insertion/deletion differences from other Acanthamoeba sequences in the DNA databases, suggesting that the divergences were due to sequencing artifacts specific to that early study. When sequence comparisons were restricted to only the critical JDP1-JDP2 motif region of the 18S rRNA gene, 6 of the 7 sequences from Khan, Jarroll, and Paget were identical with the matched sample. Only a single sample, AF239298, representing CCAP 1501/3c (OX-1) showed differences. This last sample was at least 5% divergent from any other sequence within the DNA databases for the critical JDP1-JDP2 region and was the most divergent sample among all matched samples tested. Nevertheless, despite questions about the sequence, the placement of AF239298 within the phylogeny of Acanthamoeba is still within the super-type T2/T6, consistent with the placement of the other sample of CCAP 1501/3c (Genbank AF019051).

In determining what sequence were included in our analysis, we considered one additional very early set of sequences in the DNA databases, among the earliest attempts to identify Acanthamoeba isolates by DNA sequencing [42]. Five standard strains were among the isolates examined in that report and the sequences are illustrative of early problems with manual DNA sequencing. Although a BLAST search using these sequences as the query usually indicates that they represent Acanthamoeba, and usually places the sequence within the Sequence Type of the true standard strain, the comparisons show ~80% or lower sequence similarity with their later homologues in the databases. Many sites within the gene are recorded as ambiguous, and there are numerous insertions or deletions indicating difficulty in reading the sequences manually. All of the sequences from that study have been eliminated from further consideration here.

One final consideration is necessary to identify possible sources or error. This involves the fact that some proportion of Acanthamoeba isolates for which 18S rRNA gene sequences have been obtained may be carrying more than one allele for the gene [12,43]. The ability to read the 18S rRNA gene sequence often deteriorates within a region that we have defined for alleles [12,44]. Careful analysis of the sequences electropherogram often permits the sequences of multiple alleles to be teased apart. How many copies of the 18S rRNA gene are carried within the genome of an amoebae? Recent use of long sequencing technologies on Acanthamoeba indicates that in some isolates the number is exactly two copies, as part of a tandem repeat of the ribosomal RNA gene region [45]. The sequence of the two copies of the 18S rRNA gene from the Neff isolate differed by a single nucleotide, a difference the result of length in a mononucleotide repeats within the gene [45]. In contrast, the two 18S rRNA gene sequences from the C3 isolate (ATCC 50739) differed by 17 nucleotide changes [45]. The C3 alleles can be placed into two distinct allele classes. Similar duplicated/divergent alleles have been observed by examining the genome sequences of A. culbertsoni (ATCC 30171), A. castellanii Ma (ATCC 50370), A. polyphaga JAC/S2 (ATCC 50372), and A. sp. Galka (ATCC 50496). Given that these copies within a genome are somewhat divergent, at least 5 of the pairwise comparisons in Table 1C that show similarities under 99% actually involve comparison of divergent alleles within an isolate, not either sequencing error or misidentification of the isolate.

In summary, sequence comparisons of matched samples of the 18S rRNA gene sequences from different laboratories are reproducible with less than 1% error in over 92% of appropriate comparisons, with only about 2% of comparisons showing more than 2% differences. When putative sequencing artifacts and mislabeling are subtracted, our results indicate that sequence similarities are in excess of 99.5% for most paired comparisons. Nevertheless, the results also clearly warn that care must be taken to ensure the accuracy of sequences being deposited in the DNA databases. The availability of a standardized alignment of almost complete sequences that are considered accurate would allow researchers to compare their results and evaluate any differences that are being proposed. To help researchers to quality control their results, we have provided alignments for each of the Sequence Types at http://u.osu.edu/Acanthamoeba, accessed on 20 December 2022.

### 5.3. Reliability of Mitochondrial 16S-like Sequences in the DNA Database

Questions about accuracy or error of the mitochondrial ribosomal small subunit rRNA gene of Acanthamoeba are very straightforward, compared to questions raised for the nuclear 18S rRNA genes. Almost all of the 16S-like rRNA gene sequences come from two studies centered on the gene and from genomic studies of individual isolates. The mitochondrial gene has been studied in only 49 of the 190 isolates maintained by culture centers. As mentioned, only 12 strains have been studied multiple times. Consistency of the 16S-like rRNA sequences was greater than that seen in the study of 18S rRNA gene sequences. Among the 25 pairwise comparisons, only two yielded sequence similarities were below 99.65%, as seen in Figure 2.

### 5.4. rRNA Sequences for Other Acanthamoebidae

The culture centers have maintained isolates of both Balamuthia mandrillas and Protacanthamoeba caledonica. Appendix A lists information for strains of Balamuthia and Protacanthamoeba. For the three B. mandrillaris cultures maintained by ATCC, two have been examined using both 18S rRNA and mitochondrial 16S-like sequences (Appendix A). The single Protacanthamoeba culture from ATCC has been studied only for the sequence of the 18S rRNA gene (Appendix A). Five isolates of B. mandrillaris are available through BEI. One BEI isolate has been studied for both SSU rRNA genes, three have been studied only for the mitochondrial gene, and the remaining isolate has no genetic information available. Among all of these isolates, only a single isolate (ATCC 50209) has been studied more than once independently, having three 18S rRNA gene sequences. Pairwise comparisons among the three sequences exceed 0.995.

### 5.5. Phylogenetic Implications of rRNA Sequences from Culture Center Strains

Comparisons of the phylogenetic position derived from independent analysis of the two rRNA genes has been shown to reveal some discrepancies in the placement of groups with *Acanthamoeba* [13]. The placement of the T5 Sequence Type, represented by *A. lenticulata*, is different between the two genes. Detailed placement among the subtypes of the T4 Sequence Type also shows some differences, especially in the ability of the mitochondrial 16S-like rRNA gene to separate T4 subtypes T4A and T4B. This is only to be expected of genes that have (1) very different rates of change within the phylogeny, and (2) possibly different patterns of inheritance, although patterns of inheritance for nuclear compared to mitochondrial genes in *Acanthamoeba* has not been determined definitively. Comparisons of the sequences from entire mitochondrial genomes suggests that most of the discrepancies of T4 sequence subtypes disappear, although some mixing of sequence subtypes T4A and T4B may still exist (unpublished). The use of additional loci is clearly an approach to be taken in the future. Complete genome sequences will provide the ability to examine how many genes evolve within free-living amoebae. Multilocus sequence typing (MLST) is certainly an approach [46], especially if more genomic information is forthcoming on various standard strains.

At the present time there have been few loci other than the two small subunit rRNA genes for which a substantial number of isolates have been studied. Among the few regions with more than 20 sequences from various isolates are portions of the mitochondrial cytochrome C oxidase subunit I/II gene [47,48,49,50], the mitochondrial ND5 gene [47], the large subunit rRNA and the ITS region of the ribosomal RNA complex [51], the mannose and laminin binding proteins [52], and a set of 5 proteins: Beta Tubulin (*BETA*), Glycogen Phosphorylase (*GLYP*), Glyceraldehyde-3-phosphate dehydrogenase (*G3PD*), Elongation Factor 1 alpha (*ELF1*), and *RASC* [53].

## 6. Comparing Standard Strains to Isolates from Clinical or Environmental Studies?

Since our first description of a molecular typing system for *Acanthamoeba* based on the nuclear 18S rRNA gene [28], more than 6000 isolates of *Acanthamoeba* have been examined using DNA sequencing of the nuclear 18S rRNA gene [29]. Details of the phylogeny of *Acanthamoeba* continue to be revealed as additional information is added to the DNA databases [12,14,54,55,56]. Given the importance of the standard strains in shaping our approach to the study of *Acanthamoeba*, several aspects of the sequences from standard strains have been examined.

### 6.1. What Proportion of the DNA Database Is Made up of Sequences Determined from the Standard Strains?

By examining the data provided in Table 1, information from sequences has been complied for 151 standard strains. Some strains have both rRNA genes sequenced, while other strains have been sequenced multiple times for one or both genes. The strains have yielded 215 18S rRNA gene sequences from the databases, representing and 73 mitochondrial 16S-like rRNA gene sequences. These two samples represent greatly different proportions of the overall rRNA databases for *Acanthamoeba*. The 73 mitochondrial sequences 36% of the 203 *rns* sequences in our DNA database. In contrast, the 215 nuclear 18S rRNA gene sequences from standard strains represent only 3.4% of the more than 6000 *Rns* sequences in the DNA database. The 151 strains represent less than 2% of the isolates of *Acanthamoeba* that have been studied, a slightly smaller percentage than sequences, since standard strains account for many of the isolates that have been studied for both genes and account for most of the isolates in which multiple sequences of the same gene have been deposited in the databases.

### 6.2. How Does the Classification of Standard Strains Mirror the Occurrence of Sequence Types in Samples Isolated in Clinical or Environmental Studies

We compared the distribution of Sequence Types as originally defined by our lab [26,28], and as it has evolved over the intervening years [12,13,14,29,54,56,57,58,59,60,61]. The Sequence Types assigned to an isolate are determined by examination of the sequences of the nuclear 18S rRNA gene. The frequencies of Sequence Types of culture center strains examined in this study were compared with the strains obtained in surveys and subsequently deposited in the DNA databases. The results are presented in Table 3.

The distributions show considerable superficial similarities, although differences in the frequencies of types between the data on standard strains and the data from other clinical or environmental isolates result are statistically significantly different when tested by a goodness of fit test (combining some sequence classes to adjust for small class sizes in the standard strain data; *p* < 0.01, chi square > 40 with 14 d.f.).

Seven of the Sequence Types or subtypes were not found to be held in any of the culture centers. All of these absent Sequence Types are rare, with only a single type, T20, occurring in more than 0.05% of survey results.

The T4 Sequence Type is slightly under-represented in the culture center strains, although among T4 subtypes the T4-neff (or T4G [13]) subtype is over-represented in the culture center strains. This is because, even though Neff-like sequences represent only about 5% of all Acanthamoeba isolates, the A. castellanii Neff strain itself has been repeatedly resequenced by different investigators. As mentioned, the Neff strain occurs in at least two different versions within the ATCC cultures (ATCC 30010 and ATCC 50373) and in the CCAP records (CCAP 1501/1a and CCAP 1501/1b). Various versions of the Neff strain are undoubtedly the most utilized strain in physiological studies of Acanthamoeba.

Among other Sequence Types, the T5 Sequence Type is the only type that appears to be substantially over-represented in the culture centers. In contrast, the T15 Sequence Type is underrepresented, primarily because only a single standard strain, ATCC 30732, has been deposited.

## 7. The Relationship between Species Name, Species Type Sample, and Genotypic Information

Standard strains are among the most important resources available to help define the identity and extent of species within *Acanthamoeba*. For many years, as biochemical and molecular data have been accumulating, questions have been raised concerning the validity of species designations in *Acanthamoeba* [12,13,25,26,27,55,62,63,64,65]. Standard strains are especially important in the study of *Acanthamoeba* because a number of them represent the type samples for various nominal species in the genus. Many species names have been proposed for members of what has become the genus *Acanthamoeba,* beginning even before the proposal of Volkonsky [66] for a new genus, *Acanthamoeba*, within his proposed new subfamily Hartmanellinae.

Over the last century, at least 30 species names have been applied to strains that would be considered to be within the current genus *Acanthamoeba*. A number of other names were associated with reports from early in the 20th century for which no surviving culture or biological material remains. The beginnings of a modern description of the genus *Acanthamoeba* can be traced to Volkonsky [66]. Major reinterpretations of species designations and relationships were made by Page [24] and Pussard and Pons [67]. Additional studies have continued, with the latest formal proposal of a new species occurring in 2021 [14].

The analysis of questions concerning the relationship between phylogenetic similarities, assessed using DNA sequences, and a system of species identifications using accepted binomial species names depends on our ability to examine how traditional criteria that resulted in species identification compare with newer molecular techniques of classification. The standard strains maintained by the culture centers represent a set of strains for which traditional methods of classification have been an important element in the description of a strain. In fact, the standard strains may represent the definitive set of strains for testing the definition of species in *Acanthamoeba*.

### 7.1. Type Isolates Maintained in the Culture Centers—A. castellanii versus Neff

The type specimen for the genus *Acanthamoeba* and for the species *A. castellanii* continues to be available for study. This is the strain sometimes referred to as Ac30 and listed as ATCC 30011 or CCAP 1501/2a (although there also exist several subcultures of the same strain) in both culture center inventory. Both nuclear and mitochondrial small subunit rRNA sequences have been obtained for this strain. The sequences place this strain within Sequence Type T4, and in subtype T4A [29]. It should be noted that Pussard and Pons (1967) list the type strain of *A. castellanii* as the Neff strain, which does not appear to be the case. However, based on precedent of dates of collection and on the description by Volkonsky [66], the Neff strain is clearly not the type strain for *A. castellanii*. Although the Neff strain is found to be within Sequence Type T4, on the basis of sequences from both nuclear and mitochondrial rRNA genes, the Neff strain is classified into a very different sub-type of genotype T4, designated as T4-neff [29]. Corsaro [13] recently noted that the Neff strain shares 100% sequence similarity for the 16S-like rRNA gene with the strain ATCC 30134 *A. terricola* and suggests that the Neff strain be designated *A. terricola* Neff and be that the T4-Neff subtype be re-designated T4G, a suggestion that we support.

### 7.2. Type Isolates Maintained in the Culture Centers—Other Species

How many type strains representing the other named species of *Acanthamoeba* are available from the culture collections, and for how many of these strains do we have sequence information? The list of species names, together with culture collection identification numbers, and information concerning the molecular data that are available is presented in Table 4, with the named species separated into morphological groups as defined by Pussard and Pons [67]. Of the accepted species in *Acanthamoeba*, 27 have at least a putative type strains that exist in the culture centers. The same is true for two additional proposed species. Some sequence data can be used to examine the acceptability of species designations for all 29 of these standard strains. The data for the type strains are presented in Table 4.

If we make an assumption that Sequence Type or significant subtype is equivalent to a molecular species [65], then we would expect that no more than a single type strain for the nominal species should be found in each Sequence Type or subtype. That is not the case for five of the classes of Sequence Types or subtypes. Sequence type T4D contains 6 different type strains for nominal species, Sequence Type T4A contains 3 type strains of nominal species, and Sequence Types T3, T6, and T11 each have 2 type strains of nominal species.

An alternative approach to study how standard strains impact our view of species in *Acanthamoeba* is to examine whether more than one Sequence Type designation is found associated with a particular species name, i.e., if that name has been assigned to multiple standard strains. The standard strains assigned to 8 species names are found to be allocated to more than one Sequence Type or sub-type. These species names are listed in Table 5, along with the Sequence Type or sub-type into which their sequences are assigned.

Table 5 shows that multiple Sequence Types have been associated with the same species name and multiple species have been associated with the same Sequence Type. It is clear that species designations, as currently applied, will lead to ambiguity in classification. If Sequence Types are viewed as the most accurate classification of phylogenetic relationships between strains of *Acanthamoeba*, then numerous synonymous names exist. There are also a number of Sequence Types or sub-types that do not have a species name associated with them among the standard strains. There are further Sequence Types that do not have a standard strain in the culture collections.

One final anomaly of the data must be mentioned. In general, assignment to Sequence Type of species and morphological group are consistent (Table 4). There are some inconsistencies, especially for the species *A. echinulata*. This species has been assigned to morphological group I [67]. Electron microscopic analysis of the type strain of *A. echinulata* was consistent with a placement within group 1 *Acanthamoeba* [68]. Protein analysis indicated that *A. echinulata* and *A. comandoni* have the same isoenzyme pattern [69]. However, the single partial 18S rRNA gene sequence that is available would contradict this similarity. The DNA sequence would place the standard strain (ATCC 50239) that was being analyzed, incidentally by the author’s own laboratory [70], into Sequence Type T4D, which is expected to be a member of morphological group 2. It is unclear what the status of ATCC 50239 is in the culture center, since no other analysis has been performed. This is likely to represent another case of mislabeling and needs to be examined to see whether the culture available from ATCC is correctly a member of morphological group I, and what its classification would be according to DNA sequences.

## 8. Discussion

*Acanthamoeba* is a genus that is very widespread, and whose members appear capable of surviving under a very wide variety of environmental conditions. The genus is also characterized by its ability to harbor, and even foster, a variety of pathogenic bacteria, making Acanthamoebae potential vectors for many “non-amoebic” diseases [21,22,71,72,73,74,75,76]. Do genetic differences between different genotypes of *Acanthamoeba* play an important part in the ability of an individual *Acanthamoeba* to carry a specific array of bacteria? The answer is currently unknown. Knowledge about the genetic relationships among standard strains of *Acanthamoeba* is important in guiding our approach to research in this area.

Standard strains of microbial organisms are extremely important as a resource for the conduct of biological and microbial research. However, in this molecular era of research, the utility of standard strains maintained by stock culture collections is restricted if data concerning their molecular identification are missing from the inventory of background information of the strain. In the case of standard strains of the Acanthamoebidae, almost 80% of standard strains can be classified currently using at least the nuclear 18S rRNA gene. Those standard strains that have been classified roughly represent the diversity of isolates that have been reported in the combination of environmental and clinical studies of *Acanthamoeba*. However, there remain almost 40 isolates for which little information is available.

Our knowledge of strain interrelationships is an important factor in guiding future approaches to genomic analysis of *Acanthamoeba*. Current information concerning the genome of *Acanthamoeba* comes from analysis of a small proportion of the standard strains. Genome information has long been dominated by insights provided by analysis of the Neff strain. Although the Neff strain has traditionally been viewed as descriptive of the species *A*. *castellanii*. this strain may actually be quite unrepresentative of *Acanthamoeba* and is unrepresentative of even the T4 Sequence Type of *Acanthamoeba* [29]. Genome information is currently available from only 8 of the 23 Sequence Types within *Acanthamoeba* (T2, T3, T4, T5, T7, T10, T18, T21, and T22). We do not even know for certain what strain or species is represented by the T22 genome sequence. Even within T4, genome information is only available for 5 of the 8 subtypes (T4A, T4B, T4D, T4F, and T4G). It should be a priority to obtain genomic information from a more diverse and representative set of strains.

Above, we mentioned that multilocus analysis of strains may be a way to better understand the biology of *Acanthamoeba*. MLST analysis has been used to provide significant insights into the interrelationships among groups at various taxonomic levels of the Amoebozoa [56,77,78,79,80]. Using datasets that include from 187 [77] to 1559 [79] genes, the relationships among major groups within the Amoebozoa is being scrutinized. Members of the Acanthamoebidae have been a focus of some of these studies [56,77,80]. The increasing number of genomic studies on strains of *Acanthamoeba* make it likely that multilocus analyses will soon be used to more precisely examine relationships within the genus. Understanding the value of the strains maintained by the culture centers and building on the foundation of information from analysis of the small subunit rRNA genes must be used as a guide to these future studies.

There are parts of the biological diversity of the Acanthamoebidae that are underrepresented by standard strains. This is especially true for many of the newer genotypic Sequence Types that have been identified in the last ten years, for instance T13, T14, T16, T19, and T20. These Sequence Types appear to be less prevalent in the environment than many of the Sequence Types that were first identified. It is certainly likely that additional rare Sequence Types will be discovered in the future, and these will also not be represented in the current collections. Again, even in the T4 Sequence Type, a number of divergent isolates appear to be present, with many being assigned to T4E simply because of their divergence. Almost none of these strains have been represented by even a complete 18S rRNA gene sequence, being glimpsed only through partial sequences. Much remains to be done.

The portion of the genus *Acanthamoeba* that is represented by the members of morphological group I, which possess larger trophozoites and cysts than other types, may also be under-represented in the culture collections. The DNA sequences for group I *Acanthamoeba* in the DNA databases suggest that this part of the genus is very diverse. While representing only a small proportion of the isolates whose sequences have been deposited in the DNA databases, comprising only about 1.5% of all 18S rRNA sequences in the *Acanthamoeba* database, group I forms represent at least 5 of the 23 Sequence Types that have been described. Clinical studies indicate that group I forms are less likely to be recovered in a clinical setting, and less emphasis has been placed on identifying the environmental conditions that favor the recovery of group I forms. Given the small sample size of isolates from *Acanthamoeba* group I, it is very likely that in the future additional Sequence Types (or species) may be identified from this part of the genus. This represents an area in which further effort should be placed on ascertaining the molecular differences between group I forms and other members of *Acanthamoeba*. The degree of genetic differentiation of the ribosomal RNA genes of morphological group I from other *Acanthamoeba* is likely to be mirrored in genes associated with adaptation to a different ecological niche. They may well form important contrasts as we move to better understand the genetics of ecological adaptation in amoebae.

It would have been interesting to contrast the frequency with which species names are applied to standard strains in comparison with the frequency that they are applied to isolates from clinical or environmental surveys, as well as to examine the distribution of species names with respect to Sequence Types in survey material. However, this is likely to be a deceptive comparison. Application of a species names to a standard strain is likely to have occurred because the strain had been examined morphologically and the morphological analysis indicated that the strain fits criterion laid out in keys such as Page [81,82] or Pussard and Pons [67]. In contrast, when a species name is applied to an isolate in a clinical survey during which DNA sequence information was collected, or especially when applied to an isolate in a similar environmental survey, there is a reasonable chance that the application of the species name was made because the sample showed high sequence similarity to a previous sequence present in the international DNA databases from an isolate to which a species name was assigned. Although some surveys do include information concerning morphological characters, such as trophozoite size and cyst morphology, most surveys do not include such an examination of the amoebae.

In the future, the culture collections should be encouraged to assist in completing the molecular inventory of strains, both by obtaining sequences for those strains that have been unstudied, and by making sure that nearly complete sequences for both rRNA genes have been obtained. It should be the responsibility of researchers in the *Acanthamoeba* community to ensure that strains representative of genotypes and Sequence Types missing from the culture collection are provided to the culture centers. Given the potential importance of *Acanthamoeba* as a health threat, the collaboration of culture centers with the international community of researchers has the potential to serve as a model of the scientific analysis of lower eukaryotes.

## Figures and Tables

**Figure 1 microorganisms-11-00295-f001:**
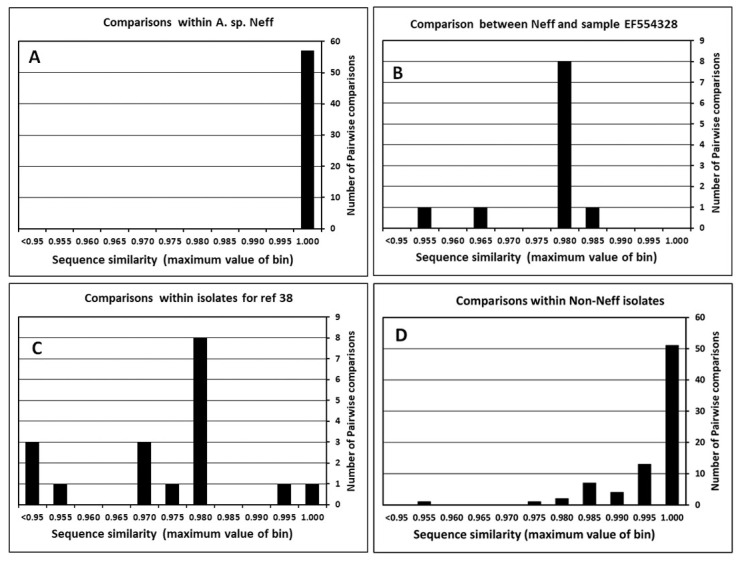
Pairwise sequence similarity for multiple sequences of the same 18S rRNA gene within an ATCC isolate: (**A**) Pairwise comparisons between independent determinations of the 18S rRNA genes of A. castellanii Neff; (**B**) Pairwise sequence similarity between various 18S rRNA gene sequences from Neff isolates and the putative Neff isolate represented by GenBank acc # EF554328; (**C**) Pairwise sequence similarity when comparing sequences from [41] with equivalent sequences from the same ATCC/CCAP isolates determined in other studies; (**D**) Pairwise sequence similarities for sequences of the 18S rRNA genes within an ATCC/CCAP isolate with multiple sequence determination, excluding the results from Figure 1A–C.

**Figure 2 microorganisms-11-00295-f002:**
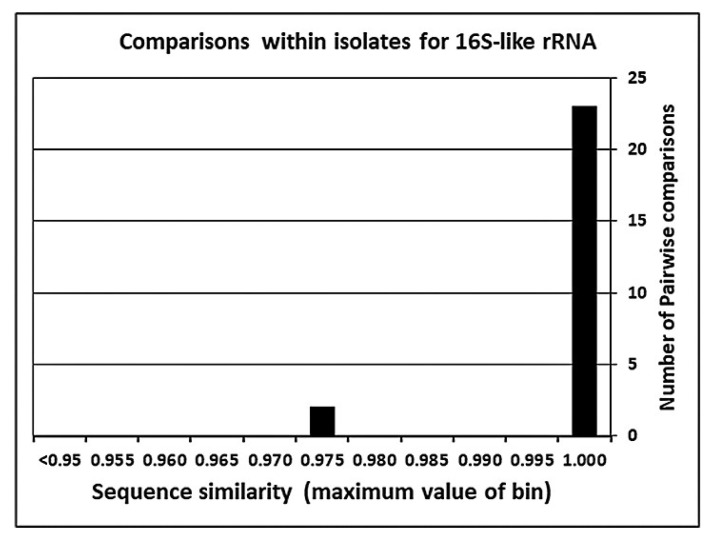
Pairwise sequence similarity between multiple independent sequences of the mitochondrial 16S-like rRNA gene within an ATCC isolate.

**Table 1 microorganisms-11-00295-t001:** Summary of standard strain *Acanthamoeba* cultures available from the culture centers or discontinued, as analyzed using molecular methods.

Types of Molecular Data Collected for an Isolate	Cultures Available from ATCC	Available in BEI (Not in ATCC)	Available in CCAP (Not in ATCC)	Discontinued or Unavailable from ATCC or BEI or CCAP	Totals
Both 18S and 16S-like rRNA	48	0	3	1	52
only 18S rRNA	79	9	4	5	97
only 16S-like rRNA	2	0	0	0	2
only allozymes and/or RFLP	14	0	0	0	14
no molecular studies	20	0	1	4	25
total	163	9	8	10	190

**Table 2 microorganisms-11-00295-t002:** Culture center strains that have been studied in genomic surveys.

ATCC/CCAP ID	Strain/Species Name	Whole Genome Shotgun Project (WGS)
ATCC 30010		AEYA
	*A. castellanii* Neff	AHJI
		JAJGAP
ATCC 30137	*A. astronyxis*	CDFH ^1^
ATCC 30171	*A. culbertsoni strain A1*	CDFF
ATCC 30461	*A. polyphaga*	SRX18334599
ATCC 30841	*A. lenticulata* isolate PD2S	CDFG
ATCC 30870	*A. palestinensis* Reich	CDFA ^2^
ATCC 30872	*A*. sp. strain Page 45	CDFK ^3^
ATCC 30973	*A. rhysodes* isolate Singh	CDFC
ATCC 50240	*A. lugdenensis* isolate L3a	CDFB
ATCC 50241	*A. quina* isolate Vil3	CDFN
ATCC 50253	*A. mauritaniensis* isolate 1652	CDFE
ATCC 50254	*A. triangularis* isolate SH621	CDFD ^4^ CACVKS
ATCC 50370	*A. castellanii* (isolate Ma)	CDFL
ATCC 50496	*A. sp.* (strain Galka)	CDFJ ^5^
ATCC 50704	*A. lenticulata* strain 72/2	MSTW
ATCC 50739	*A. castellanii* strain C3	JAJGAO
ATCC PRA-287	*A. comandoni* strain Pb30/40	MRZZ
CCAP 1501/18	*A. polyphaga* strain Linc Ap-1	LQHA
CCAP 1501/19	*A. pyriformis* CR15	SRA SRX2163158

^1^—genome project, CDFI, putatively examining A. divionensis ATCC 50238 was actually a duplicate analysis of A. astronyxis ATCC 30137. ^2^—mislabeled as A. healyi. ^3^—conflict in sequence with AY026244. ^4^—mislabeled as A. palestinesis. ^5^—putative assignment based on best match—erroneous listing as A. pearcie ATCC 50436.

**Table 3 microorganisms-11-00295-t003:** Frequency of Sequence Types for standard strains with genotype data and percentage of Sequence Types in all isolates from clinical or environmental surveys.

Sequence Type	#	% Standard Strains	% Survey Strains
T1	2	1.32%	0.43%
T2	3	1.98%	2.20%
T2/6a	2	1.32%	0.65%
T2/6b	0	0.00%	0.43%
T2/6c	1	0.66%	1.72%
T3	5	3.31%	5.60%
T4 (total)	98	64.90%	70.30%
T4A	42	27.81%	25.52%
T4B	18	11.92%	12.59%
T4C	5	3.31%	7.11%
T4D	12	8.6%	10.60%
T4E	13	9.62%	6.08%
T4F	2	1.32%	2.23%
T4-neff	6	7.05%	4.98%
T5	18	11.92%	6.98%
T6	4	2.65%	1.75%
T7	1	0.66%	0.18%
T8	1	0.66%	0.03%
T9	2	1.32%	0.44%
T10	1	0.66%	0.32%
T11	3	1.92%	1.91%
T12	2	0.64%	0.43%
T13	2	0.64%	1.06%
T14	0	0.00%	0.08%
T15	1	0.64%	2.60%
T16	0	0.00%	0.35%
T17	0	0.00%	0.31%
T18	1	0.64%	0.50%
T19	1	0.64%	0.10%
T20	0	0.00%	0.55%
T21	1	0.64%	0.02%
T22	0	0.00%	0.02%
T23	0	0.00%	0.03%

**Table 4 microorganisms-11-00295-t004:** *Acanthamoeba* species type isolates in culture centers.

Morphological Group	Species Name	ATCC Strain #	CCAP Strain #	Strain Name	Nuclear 18S rRNA Gene	Mito. 16S-like Gene	Genotype Group
Group 1	*A. astronyxis*	30137	1534/1		Y	Y	T7
	*A. byersi*	PRA-411			Y	N	T18
	*A. comandoni*	30135	1501/5		Y	Y	T9
	*A. echinulata* ^(1)^	50239		278	Y ^(1)^	N	T4D? ^(1)^
	*A. tubiashi*	30867		OC-15C	Y	Y	T8
Group 2	*A. castellanii*	30011	1501/2a	Ac30	Y	Y	T4A
	*A. divionensis*	50238		AA2	Y	Y	T4D
	*A. griffini*	30731	1501/4	S-7	Y	Y	T3
	*A. hatchetti*	30730		BH-2	Y	N	T11
	*A. lugdunensis*	50240		L3a	Y	N	T4A
	*A. mauritaniensis*	50253		1652	Y	Y	T4D
	*A. paradivionensis*	50251		AA1	Y	Y	T4D
	*A. pearcei*	50435		205-1	Y	N	T3
	*A. polyphaga*	30871	1501/3a	Page-23	Y	N	T4E
	*A. quina*	50241		Vil3	Y	N	T4A
	*A. rhysodes* ^(2)^	30973	1534/3	Singh	Y	?	T4D
	*A. stevensoni*	50388		RB-F-1	Y	N	T11
	*A. terricola*	30134			N?	Y	T4-neff
	*A. triangularis*	50254		SH 621	Y	N	T4F
	*C. operculata*	50243			Y	N	T6
	*A. sawyeri* ^(3)^	50656	NR46460	CDC:0484:V017	Y	Y	T4B
Group 3	*A. culbertsoni*	30171	1501/6	Lilly A1	Y	Y	T10
	*A. healyi*	30866		OC-3A	Y	Y	T12
	*A. jacobsi*	30732			Y	N	T15
	*A. lenticulata*	30841		PD2	Y	Y	T5
	*A. palestinensis*	30870	1547/1	Reich	Y	Y	T2
	*A. pustulosa*	50252		GE 3a	Y	N	T2
	*A. royreba*	30884		OR (Oak Ridge)	Y	Y	T4D
unknown	*A. giganteum* ^(4)^	50670		25-349-MX	Y	N	T4A

^(1)^ Sequence suggests that stock does not represent original deposit and was mislabeled at some point; sequence indicate isolate as member of Group 2. ^(2)^ Sequence conflict between ATCC 30973 and CCAP 1534/3 (ATCC 30869) [T4D]. ^(3)^ Species description never published. ^(4)^ Probable group 2 morphology.

**Table 5 microorganisms-11-00295-t005:** *Acanthamoeba* species with standard strains placed in multiple sequence types.

Species	# Sequence Types	Sequence Types
*A. astronyxis*	2	T7 T9
*A. castellanii*	6	T1 T4A T4B T4C T4D T4-neff
*A. culbertsoni*	3	T4A T4B T10
*A. hatchetti*	4	T4B T4E T6 T11
*A. mauritaniensis*	2	T4D T4E
*A. palestinensis*	3	T2 T6 T2/6C
*A. polyphaga*	6	T3 T4A T4B T4D T4E T2/6A
*A. rhysodes*	2	T4A T4D

## Data Availability

Additional details on the information in the *Acanthamoeba* DNA databases and on the analysis of species names, Sequence Types and allelic variants can be obtained on the website http://u.osu.edu/acanthamoeba/accessed on 23 December 2022.

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
