# Peer review of "The Status of Molecular Analyses of Isolates of Acanthamoeba Maintained by International Culture Collections"

_microorganisms, 2023, doi:10.3390/microorganisms11020295_

Round 1

Reviewer 1 Report

It is very important to carry out a description of the characteristics of the strains that are kept in the collections. However, this work is only a description of the way in which the specimens that make up the collections have been safeguarded. It would be worth it if it went further and could give it a better order and a better organization so that it can be used more efficiently in future studies, that a molecular analysis could be made that would allow us to be certain of the characteristics of these and thereby base future studies in any of the various fields of study.
It would be convenient to base and expand the discussion of the manuscript

Author Response

responses: 

Reviewer 1: I am not sure that I completely understand what reviewer 1 refers to.  I have added several statements in the discussion that I believe responds to the reviewer’s concerns.

Reviewer 2 Report

Manuscript ID: microorganisms-2101623

The Status of Molecular Analyses of Isolates of Acanthamoeba 2 Maintained by International Culture Collections

The present manuscript is of immense quality and represents a before and after for the knowledge of the genus Acanthamoeba. I have learnt and enjoy this wonderful review and for sure recommend it for its publication.

I have just some comments 

Figure 2 is missing

Does the author recommend to use the MLST as a new approach in order to characterize Acanthamoeba genus?

Comments

Keywords: Acanthamoeba should be in italics

What is the reason the author has included Comandonia culture in the study?

Line 130: “Additionally, there is a single standard culture of Comandonia operculatum, which has been shown genetically to be included within Acanthamoeba”. Does the author refer to Acanthamoeba genus or Acanthamoebidae family?

Line 138: It would be interesting if the author could add an example of the aim of the use of these samples.

Line 207: clarify this sentence please

Line 240: The author needs to clarify the number assigned to each table included in the supplementary data

Line 252: If the author is referring to Balamuthia and Protacanthamoeba, both should be in italics.

Line 282: eliminate the “.” after ATCC 50238

Line 292: “data”

Line 295: “Acanthamoeba” should be in italics

Line 298: “A. castellanii” should be in italics

Line 314: “A. castellanii” should be in italics

Line 315: “A. terricola” should be in italics

Line 321: “Acanthamoeba” should be in italics

Line 326: “Acanthamoeba” should be in italics

Line 332: “Acanthamoeba” should be in italics

Line 343: “Acanthamoeba” should be in italics

Line 351: “Acanthamoeba” should be in italics

Line 355: “A. castellanii” should be in italics

Line 361: eliminate the extra “.” at the end

Line 362: “Acanthamoeba” should be in italics

Line 381: “Acanthamoeba” should be in italics

Line 385: “Acanthamoeba” should be in italics

Line 388: “Acanthamoeba” should be in italics

Line 395: “Acanthamoeba” should be in italics

Line 401: “Acanthamoeba” should be in italics

Line 408: “A. castellanii”, “A. polyphaga” and “A. sp.” should be in italics

Line 426: “Acanthamoeba” should be in italics

Lines 435-445: “Balamuthia mandrillas” and “Protacanthamoeba caledonica” should be in italics

Line 449: “Acanthamoeba” should be in italics

Line 466: “, the large subunit rRNA”

Line 513: “Acanthamoeba” and “A. castellanii” should be in italics

Line 517: “Acanthamoeba” should be in italics

Line 586: Table 4: “Acanthamoeba” should be in italics

Line 640: “Acanthamoeba” should be in italics

Line 646: “A. castellanii. this strain”. This

Line 647: substitute Fuerst, 2014 by the correct number of the reference

Author Response

Reviewer 2: Thank you for the substantial set of comments.  I have carefully reviewed the paper to restore italics that appear to have been removed by the reference manager that I use (plus a few cases in the body of the paper that were missed).

Other responses:   

Line 130, concerning Comandonia. INFORMATION HAS BEEN PROVIDED.

Comment on line 138.  I HAVE ADDED SOME EXAMPLES THAT ARE RELEVANT TO THE IMPORTANCE OF THE ATCC DIFFERENTIATION BETWEEN MISSION ISOLATES AND OTHER STRAINS.

Comment on Line 207-208: SENTENCE IN QUESTION WAS A PROOFREADING NOTE THAT HAD NOT BEEN REMOVED BEFORE INITIAL SUBMISSION. 

Line 207: clarify this sentence please. DONE

Line 240: The author needs to clarify the number assigned to each table included in the
supplementary data.  MATERIAL ADDED AND CORRECTED TO CLARIFY EACH TABLE.

Line 252: If the author is referring to Balamuthia and Protacanthamoeba, both should be
in italics.   DONE. SENTENCE ALSO MOVED TO FOLLOW PATTERN FOR OTHER DESCRIPTIONS OF TABLES IN APPENDIX.

Additional comments:  Since submission, information on an additional genome project has been reported.  This information has been added and slight alteration to number in the text and tables have been made to reflect the additional data. The additional data requires no alteration in any conclusions.

Figure 1 - around line 357  - Figure 1D has been altered to add two comparisons that have become available since the initial submission.   

Table 1 altered to reflect additional data

Table 2 altered to reflect additional data

In light of the reviewers comments, an additional proofing of the manuscript was performed that resulted in movement of several paragraphs to provide a better flow for the manuscript.   

Round 2

Reviewer 1 Report

I considert hat the manuscript does not show its relevance, its discussion and conclusion are not clear, it is too superficial.
It would be important that the authors not only make a review but try to show and complete the missing information in the files of the reference strains.

Author Response

I find the second round review to be troubling. The review consists of two sentences.  I have no idea what changes this reviewer is calling for. They were equally opaque in the first round review. There is no detailing of the "missing information".  Unless I am very dull-headed, I cannot see any "missing information" except that which is "missing" from isolates that have never been studied.  I believe, as did reviewer #2, that the relevance of the material being reviewed is obvious. The reviewer provides no detail of issues that they appear to think are required to be corrected.  The contrast between reviewers #1 (here) and #2 is stark.  I do not see that any changes can be made in response to such an incomplete and undetailed review.